# Genetic drift promotes and recombination hinders speciation on holey fitness landscapes

Ata Kalirad[1,2], Christina L. Burch[3], Ricardo B. R. Azevedo[1]*

**1** Department of Biology and Biochemistry, University of Houston, Houston, Texas, United States of America, **2** Department for Integrative Evolutionary Biology, Max Planck Institute for Biology Tübingen, Tübingen, Germany, **3** Department of Biology, University of North Carolina, Chapel Hill, North Carolina, United States of America

* razevedo@uh.edu

**Data Availability Statement:** All code and data are available at https://github.com/Kalirad/HoleyPopulationSpeciation.

**Funding:** This work was funded by grants from the National Science Foundation DEB-2014566 to

## Abstract

Dobzhansky and Muller proposed a general mechanism through which microevolution, the substitution of alleles within populations, can cause the evolution of reproductive isolation between populations and, therefore, macroevolution. As allopatric populations diverge, many combinations of alleles differing between them have not been tested by natural selection and may thus be incompatible. Such genetic incompatibilities often cause low fitness in hybrids between species. Furthermore, the number of incompatibilities grows with the genetic distance between diverging populations. However, what determines the rate and pattern of accumulation of incompatibilities remains unclear. We investigate this question by simulating evolution on holey fitness landscapes on which genetic incompatibilities can be identified unambiguously. We find that genetic incompatibilities accumulate more slowly among genetically robust populations and identify two determinants of the accumulation rate: recombination rate and population size. In large populations with abundant genetic variation, recombination selects for increased genetic robustness and, consequently, incompatibilities accumulate more slowly. In small populations, genetic drift interferes with this process and promotes the accumulation of genetic incompatibilities. Our results suggest a novel mechanism by which genetic drift promotes and recombination hinders speciation.

## Author summary

As geographically isolated populations evolve, genetic incompatibilities accumulate between them. Eventually, these incompatibilities may cause the populations to become different species. What determines how quickly species are formed in this way remains unclear. We investigate this question using computer simulations and find that genetic incompatibilities accumulate more slowly among populations with individuals that are robust to genetic perturbations. We identify two factors that influence the accumulation rate via their effects on genetic robustness: the size of the populations and how much

RBRA and DEB-2014943 to CLB (https://www.nsf.gov/). The funders had no role in study design, data collection and analysis, decision to publish, or preparation of the manuscript.

**Competing interests:** The authors declare no competing interests.

recombination takes place in them. Small populations with rare recombination accumulate incompatibilities more quickly.

## Introduction

The rates of origination of new species vary extensively across the tree of life. For example, some lineages of cichlids are estimated to produce new species approximately 70× faster than some lineages of hawks [1]. A major challenge for evolutionary biology is to explain this macroevolutionary pattern through the action of microevolutionary processes operating within populations. Stanley [2] speculated that speciation is "a largely random process" and that, therefore, "macroevolution is decoupled from microevolution". There is growing support for the alternative view that macroevolution and microevolution are, in fact, mechanistically coupled (reviewed in [3, 4]). For example, Jablonski and Roy [5] found that gastropod species with broader geographic ranges had lower speciation rates, consistent with the hypothesis that high dispersal ability both promotes broader geographic ranges and hinders speciation. Here we investigate the potential roles of genetic drift and recombination in determining the rate of speciation.

Speciation results from the build up of reproductive isolation (RI) between populations. Thus, variation in the rates at which RI increases between populations as they diverge should be one of the causes of variation in speciation rates. A simple modeling framework used to study the evolution of RI is the holey fitness landscape [6]. In such a landscape, genotypes are either viable or inviable. Many types of holey fitness landscapes have been proposed. For example, Gavrilets et al. [7, 8] analyzed a model where individuals differing at a threshold number of loci were considered incapable of producing viable offspring (i.e., were reproductively isolated).

Dobzhansky and Muller proposed that, as allopatric populations diverge, genetic incompatibilities accumulate between them and generate postzygotic RI [9, 10]. Such Dobzhansky–Muller incompatibilities (DMIs) have been shown to cause low fitness in hybrids between closely related species [11–13]. A particular kind of holey fitness landscape, also known as a neutral network [14, 15], is especially suitable for modeling the evolution of RI by DMIs. A neutral network is a network of viable genotypes connected by mutational accessibility. Two genotypes are mutationally accessible if one genotype can be obtained from the other through a single mutation. Fig 1C shows a neutral network defined by five diallelic loci (yellow, orange, and purple nodes). Inviable genotypes carry combinations of alleles displaying DMIs (gray nodes in Fig 1C). If two populations evolving in allopatry have diverged at $k$ diallelic loci (Fig 1A), then up to $k − 1$ divergent alleles from one population will not have been tested by natural selection in the genetic background of the other population (see S1 Text). DMIs can be inferred by introgressing each of these $k − 1$ divergent alleles from one population into the other and counting the number of incompatible introgressions, IIs (Fig 1B).

Consider the simple holey fitness landscape (neutral network) model introduced by Gavrilets [6, 16, 17] in which genotypes are randomly and independently assigned to be either viable or inviable with probabilities $p$ and $1 − p$, respectively—dubbed the Russian roulette model. The value of the $p$ parameter determines the global structure of the neutral network [16]. If $p$ is below a critical value $p_c$, known as the percolation threshold, the neutral network is composed of many small disconnected subnetworks. A subnetwork forms a cluster of genotypes that can evolve from each other by a series of single mutations; evolution is more difficult between subnetworks. In contrast, if $p > p_c$ there is one giant subnetwork containing most viable

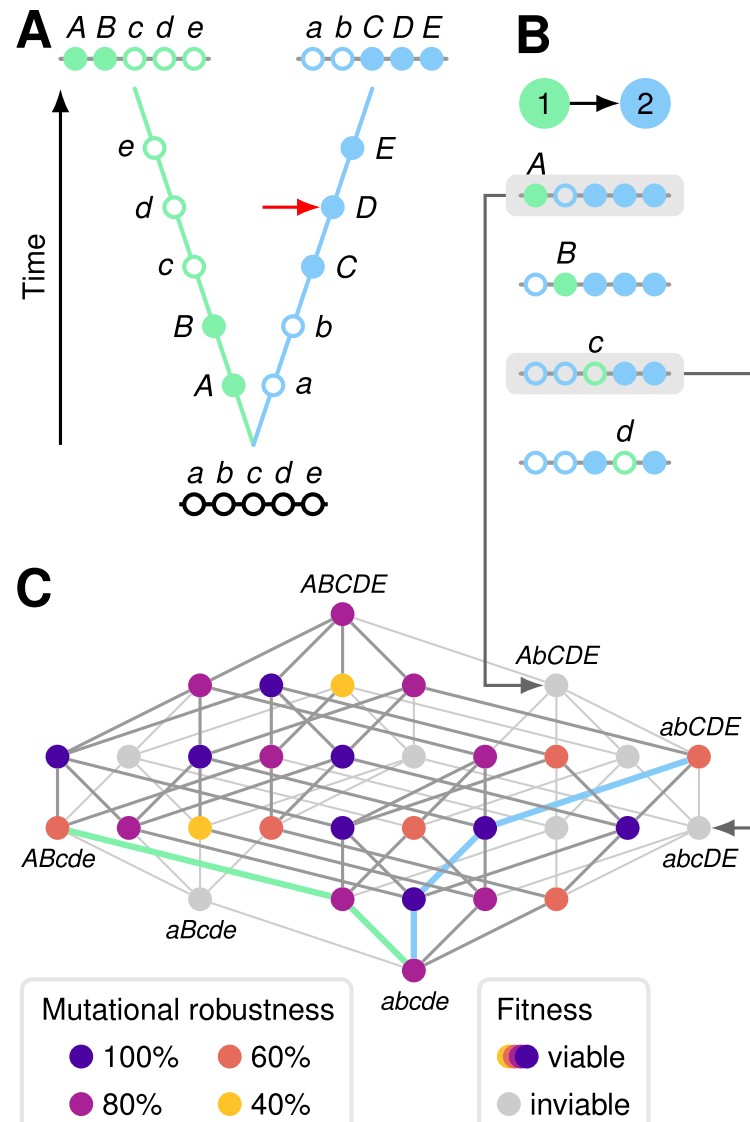

**Fig 1. Populations evolving on a holey fitness landscape accumulate genetic incompatibilities. (A)** Two populations diverge in allopatry. Both populations are initially fixed for lowercase alleles (open circles) at five diallelic loci, A–E. Derived alleles are indicated by uppercase letters (closed circles). The first two substitutions (closed circles) were of alleles *A* and *B* in population 1 (left, green); the next three substitutions were of alleles *C*, *D*, and *E* in population 2 (right, blue). **(B)** Genetic incompatibilities can be inferred by introgressing individual divergent alleles from one population into the other and counting the number of incompatible introgressions (IIs). This is illustrated by the introgression of the divergent alleles from population 1 (*ABcde*) into population 2 (*abCDE*) to create the four combinations that have not been tested by natural selection. We ignore the introgression of the *e* allele because it results in the transitional *abCDe* genotype in population 2, which is known to be viable (red arrow in (A); see S1 Text). Two alleles from population 1 (*A* and *c*) are deleterious in the genetic background of population 2 (grey boxes). We refer to these as IIs. Each II implies the existence of one or more DMIs. **(C)** Holey fitness landscape of the 32 genotypes at loci A–E. Nodes (circles) represent genotypes. Lines connect genotypes that can be reached by a mutation at a single locus. Each genotype has 5 mutational neighbors. 7 of the 32 genotypes (22%) are inviable, shown in light grey. The genotypes generated by the two IIs shown in (B) are marked by arrows. The remaining 25 genotypes (78%) are viable, shown in colors from yellow to blue, and define a neutral network. The different colors indicate the proportion *v* of the mutational neighbors of a genotype that are viable, a measure of mutational robustness. The evolutionary trajectories of the populations shown in (A) are indicated by thick blue and green lines.

genotypes. If genotypes are haploid and viability is determined by $L$ diallelic loci, the percolation threshold is $p_c \approx 1/L$.

In such a landscape, the number of IIs is expected to increase linearly with the genetic divergence between populations ($k > 0$), according to

$$\text{II}(k) = (1 - \tilde{v})(k - 1) \tag{1}$$

where $k - 1$ is the number of divergent alleles from one population that have not been tested by natural selection in the genetic background of the other population (see S1 Text for more details),

$$\tilde{v} = \frac{vL - 1}{L - 1} \ ,$$

$L$ is the number of diallelic loci affecting viability, and $v$ is the proportion of the $L$ mutational neighbors of a genotype that are viable, a measure of the genotype's mutational robustness; $\tilde{v}$ corrects for the fact that one of the mutational neighbors is *known* to be viable (see S1 Text for more details). Eq 1 indicates that the rate of accumulation of genetic incompatibilities depends on genetic robustness. This makes intuitive sense because introgression is a perturbation of the recipient genotype. Thus, microevolutionary processes that cause genetic robustness to evolve have the potential to influence the macroevolutionary rate of speciation.

If mutation is sufficiently weak that the time between the appearance of new neutral mutations ($1/(NU)$, where $N$ is the population size and $U$ is the genomic mutation rate) is much longer than the time it takes for a neutral mutation to fix ($2N$), that is, if $2N^2U \ll 1$, then evolution can be modelled as a "blind ant" random walk [18] (see Materials and methods). The expected value of $v$ in a population evolving under the Russian roulette model with Weak Mutation is simply $p$, the probability that a genotype is viable and explains the accumulation of IIs under this model (Fig 2).

The relationship between $v$ and $p$ changes when mutation is not weak. On a holey fitness landscape, all viable genotypes have the same viability but, in the presence of mutation, they do not necessarily all have the same fitness. If organisms reproduce asexually, the fitness of genotype $i$ with robustness $v_i$ is proportional to $1 - U(1 - v_i)$. Thus, if there is variation in $v$ between genotypes and mutation is not weak (i.e., if $2N^2U \gtrsim 1$), then the population will experience selection for higher mutational robustness [18] (Fig 3: $U \rightarrow v$). For example, under the Russian roulette model, genotypes show a variance in $v$ of $\text{Var}[v] = p(1 - p)/L$. Typically, recombination further strengthens selection for mutational robustness [19–21] (Fig 3: $r \rightarrow v$). The ability of populations to respond to this selection depends on factors that determine the relative strengths of selection on genetic robustness and genetic drift: mutation rate, recombination rate, and population size (Fig 3: $U, r, N \rightarrow v$) [22].

Here we use the Russian roulette model to investigate whether and why populations differ in the rate at which they accumulate genetic incompatibilities. We then validate our results using a computational model of RNA secondary structure that includes intrinsic differences in fitness between viable genotypes. We chose the folding of RNA sequences as a simple but non-random model of a genotype-phenotype map grounded in biophysical reality [23]. Crucially, this map naturally incorporates epistasis and has been used to study its evolutionary consequences (e.g., for robustness [18, 24], evolvability [25, 26], and molecular evolution [27]). Because introgression is a perturbation of the genotype, we predict that the probability that an introgression is incompatible will be inversely related to the recipient's genetic robustness, its ability to maintain viability in the face of perturbations to the genotype. We test this prediction by experimentally manipulating population genetic parameters known to promote the evolution of genetic robustness.

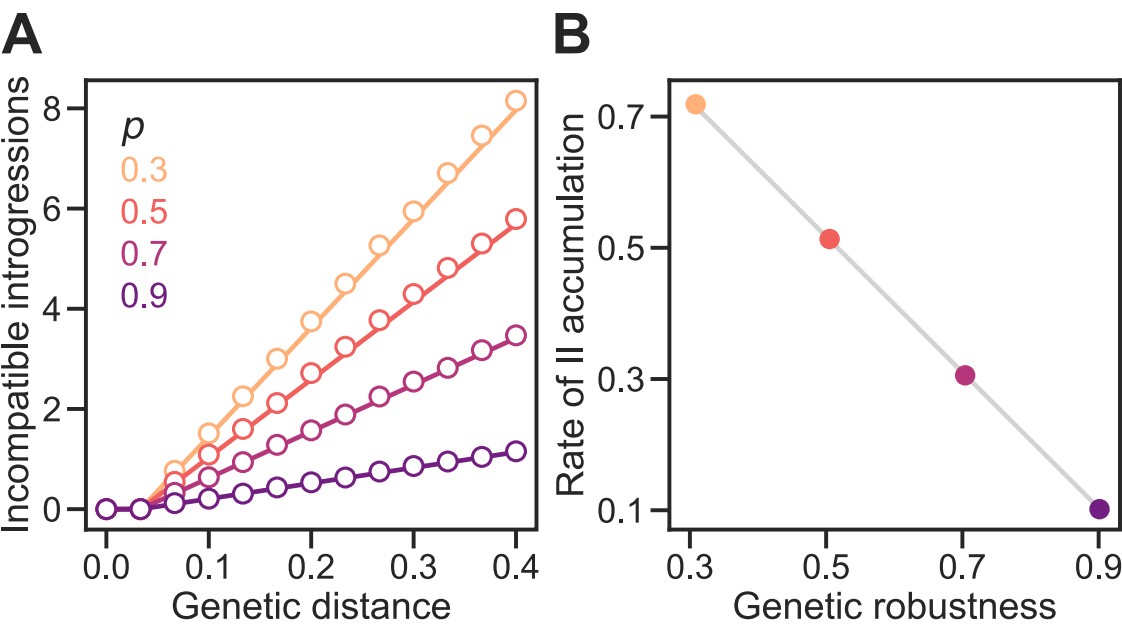

**Fig 2. Genetic robustness determines the rate of accumulation of genetic incompatibilities in a holey fitness landscape. (A)** Under the Russian roulette model incompatible introgressions (IIs) accumulate linearly with genetic distance at a rate inversely proportional to the genetic robustness of genotypes, $v$. In all simulations, populations evolved under the Weak Mutation regime, fitness was determined by $L$ = 30 diallelic loci, and genotypes were viable with probability $p$. Plotted values are means of 1,000 replicate simulation runs at each $p$ calculated for $k$ = 0, 1, . . .12 genetic differences between populations. Genetic distance was measured by $D = k/L$. In each run, the numbers of IIs and the values of $v$ of the two populations were averaged at each time step. If there were multiple observations for a given $k$, they were then averaged. Divergence was allowed to proceed until $k$ = 15 but data for $k > 12$ were discarded. The lines are the expectations according to Eq 1 with $v = p$. **(B)** We fit the linear regression model in Eq 3 to each replicate simulation when $D > 0$. Plotted values are mean values of $b/L$ and $v$ for the 1,000 simulation runs. The gray line shows $1 - \tilde{v}$ as a function of $v$ with $v = p$ (Eq 1). In both panels, the 95% CIs are hidden by the points.

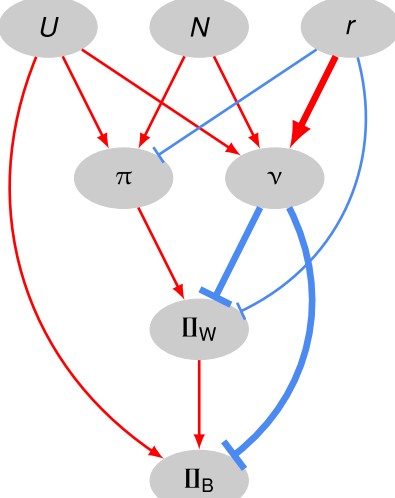

**Fig 3. Evolutionary causes of the accumulation of genetic incompatibilities between populations in our model.** Ellipses contain the following properties of populations: $U$, genomic mutation rate; $N$, population size; $r$, recombination probability or rate; $\pi$, sequence diversity; $v$, genetic robustness; $II_W$ and $II_B$, genetic incompatibilities within and between populations, respectively. Red lines ending in arrows indicate positive effects; blue lines ending in bars indicate negative effects. The thickness of the line indicates the strength of the effect.

## Materials and methods

### Experimental design

To test our prediction that the rate at which IIs accumulate between populations will be inversely related to their genetic robustness, we first experimentally evolved populations under combinations of population sizes and recombination rates expected to result in different equilibrium levels of genetic robustness. Populations evolved under an individual-based Wright-Fisher model with a selection-recombination-mutation life cycle, with a constant population size (between $N = 10$ and $N = 2000$) and recombination rate (between no recombination and free recombination). Our use of the Wright-Fisher model differs from most theoretical research into the accumulation of DMIs [28–31] in that it does not impose a Weak Mutation evolutionary regime. Our implementation of the Wright-Fisher model ensures that multiple mutations segregate simultaneously ($2N^2U \gtrsim 1$ in all of our experimental populations). This difference in approach has two important consequences for this work. It enables (1) recombination and (2) selection for genetic robustness, neither of which are effective under the Weak Mutation regime [18].

Once these populations reached an approximate mutation-recombination-selection- drift equilibrium, we made exact copies of each population to obtain pairs of identical ancestral populations. These population pairs were then allowed to diverge allopatrically, again evolving under a Wright-Fisher model with a selection-recombination-mutation life cycle and using the same population size and recombination rate used to generate the ancestral populations. We then investigated the effect that the differences in genetic robustness among the ancestral populations had on the rate at which genetic incompatibilities accumulated between the diverging population pairs.

### Genotype and phenotype

**Russian roulette model:** The genotype is a binary sequence of length $L$. In all simulations summarized in this paper we used $L = 30$ diallelic sites and assumed that genotypes are haploid. In all simulations reported in this paper we set $p \gg p_c \approx 0.033$, the percolation threshold.

**RNA folding model:** The genotype is an RNA sequence of length $L$. In all simulations summarized in this paper we used $L = 100$ nucleotides and assumed that genotypes are haploid. The phenotype associated with a particular genotype is its minimum free-energy secondary structure, as determined using the ViennaRNA package version 2.5.1 with default parameters [32].

### Selection

**Russian roulette model:** A genotype is randomly assigned to be either viable or inviable with probabilities $p$ or $1 - p$, respectively [6]. This defines a holey fitness landscape [6] or neutral network [14, 15].

**RNA folding model:** The fitness of a sequence $i$ is defined as

$$w_i = \begin{cases} 1 - \sigma\delta_i & \text{if} \quad \beta_i > \alpha \geqslant \delta_i \\ 0 & \text{otherwise} \end{cases} \tag{2}$$

where $\beta_i$ is the number of base pairs in the secondary structure of sequence $i$, $\delta_i$ is the base-pair distance between the secondary structure of sequence $i$ and a reference secondary structure, $\sigma$ is the strength of selection against $\delta_i$, and $\alpha$ is an arbitrary threshold. This means that the population experiences direct selection to resemble the reference (optimal) structure and indirect selection to stay away from the holes (defined by $\alpha$). In all simulations summarized in the main text of this paper we used $\sigma = 0.025$ and $\alpha = 12$.

Kalirad and Azevedo [31] investigated an RNA folding model without intrinsic differences in fitness between genotypes ($\sigma = 0$) in the Weak Mutation regime. They found that changes to $\alpha$ had only small effects on the mean genetic robustness of genotypes [31]. Genetic robustness in the RNA folding model does not depend on $\sigma$. We present the results of simulations on the model with $\sigma = 0$ in S1 and S2 Figs.

## Recombination

From a population of size $N$, we randomly sample with replacement two sets of $N$ viable genotypes, $S_1$ and $S_2$. Genotypes from $S_1$ and $S_2$ are paired, such that the $i$th genotypes from each set recombine.

**Free recombination:** In the free recombination regime, every offspring is recombinant. Crossovers occur independently with probability 0.5 at each of the $L - 1$ positions between sites in the genome.

**Limited recombination:** In the limited (not free) recombination regime, offspring are recombinant with probability $r$ resulting from a single crossover located at random at one of the $L - 1$ positions between sites in the genome.

**No recombination:** If there is no recombination, the offspring is a copy of a randomly selected parent.

## Mutation

Mutations arise according to a binomial process where each site mutates with probability equal to the per-site mutation rate $\mu$ in each generation. All types of point mutations occur with equal probability. Insertions and deletions are not considered. In most simulations summarized in this paper we used a genome-wide mutation rate of $U = \mu L = 0.1$.

## Ancestor

**Russian roulette model:** We generate a random viable genotype with $L = 30$ and define it as the ancestor.

**RNA folding model:** We generate a random RNA sequence with $L = 100$ nucleotides and define its minimum free-energy secondary structure as the reference, provided it is viable ($\beta_i > \alpha$ in Eq 2). We define this sequence as the ancestor.

## Evolution

**Weak Mutation simulations:** Evolution is modelled as a "blind ant" random walk [18]. The population is represented by a single genotype. At the next time step, one of its $n$ mutational neighbors is chosen at random. If the mutant is viable, the population "moves" to the mutant genotype; otherwise, the population remains at the current genotype for another time step. The genetic divergence between populations, $k$, can decrease as well as increase.

**Wright-Fisher simulations:** Initially, the ancestor is cloned $N$ times to create a population. We allow this population to evolve for as long as it takes to reach an approximate mutation-recombination-selection-drift equilibrium for mutational robustness and average sequence diversity (evaluated based on the average of 200 replicate populations). We then make an exact copy of the resulting population to obtain two identical ancestral populations. These populations are allowed to evolve under a Wright-Fisher model following a selection-recombination-mutation life cycle for as long as it takes to reach a predetermined level of genetic distance (measured using Jost's $D$ [33]).

## Incompatible introgressions

Two viable genotypes, $i$ and $j$, differ at $k$ sites. The number of incompatible introgressions [31] (IIs) from genotype $i$ to genotype $j$, $II_{ij}$, was measured by introgressing each allele at a divergent site from genotype $i$ to genotype $j$, one at a time, and counting the number of inviable introgressed genotypes ($w = 0$, see Eq 2). $II_{ij}$ is not necessarily equal to $II_{ji}$. As populations diverged, we periodically estimated $II_{ij}$ in a particular direction (e.g., from population 1 to population 2; Fig 3: $II_B$) from 100 (or $N$ when $N < 100$) randomly chosen pairs of individuals from the two populations, sampled without replacement.

## Rate of II accumulation

We fit the following linear regression model to each replicate simulation:

$$II = a + bD \tag{3}$$

where $D$ is the genetic distance between populations ($k/L$ in Fig 2A or Jost's $D$ [33] in Figs 4A and 5A, etc). The II counts at the start of divergence ($D = 0$) were not considered in the regressions (see Eq 1). The rates of II accumulation summarized in Figs 2B, 4B and 5B, etc, were calculated by averaging estimates of $b$ from individual replicate runs.

## Segregating IIs

We also counted IIs within populations as they diverged (Fig 3: $II_W$). We used the same approach as for IIs between populations, except that the 100 (or $N$ when $N < 100$) pairs of individuals were sampled randomly without replacement from the same population.

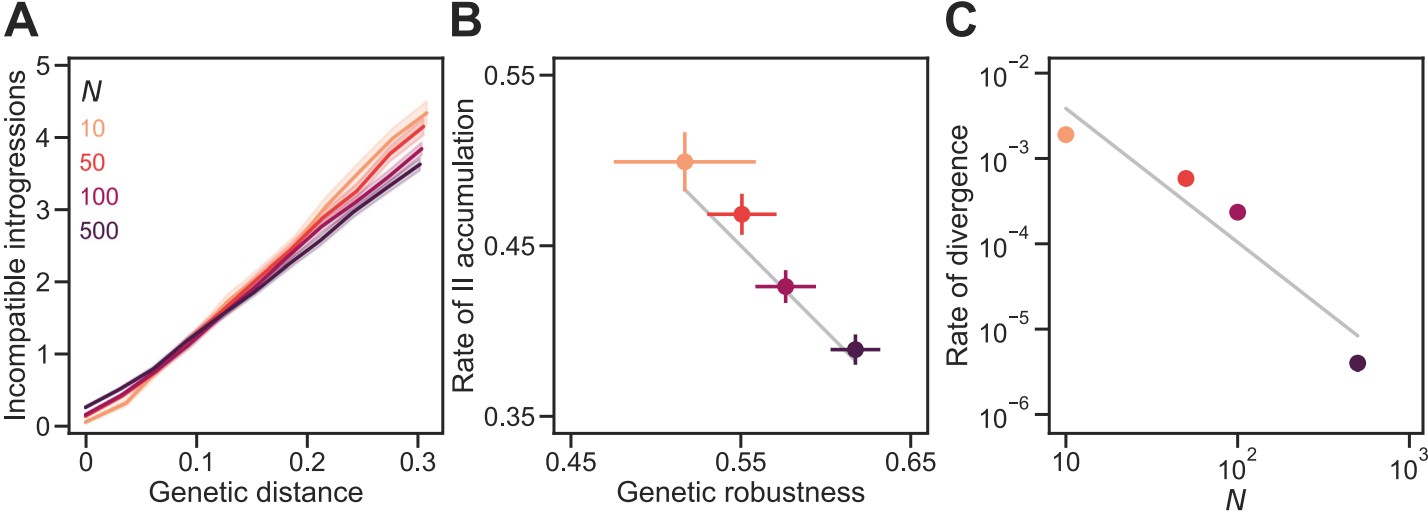

**Fig 4. Genetic drift promotes the accumulation of genetic incompatibilities. (A)** Incompatible introgressions (IIs) accumulated linearly with genetic distance in sexual populations of different sizes ($N$) evolving under the Russian roulette model. Genetic distance between divergent populations was measured by Jost's $D$ [33]. In all simulations, the proportion of viable genotypes over the entire holey fitness landscape was $p = 0.5$, the genome length was $L = 30$ sites, and the mutation rate was $U = 0.1$ per genome per generation; populations experienced random mating and free recombination. **(B)** IIs accumulated faster in smaller populations because they evolved lower genetic robustness, $v$. We fit the linear regression model in Eq 3 to each replicate simulation when $D > 0$. Plotted values are mean values of $b/L$ and $v$. The gray line shows $b/L = 1 - v$ (see Eq 1). **(C)** Large populations diverged more slowly. Values show the rate of increase in $D$ per generation. The gray line shows a power law with exponent $-1.6$. Plotted values in all panels are means of 200 replicate simulation runs at each $N$. Error bands in (A) and bars in (B) are 95% CIs (in (C) they are hidden by the points).

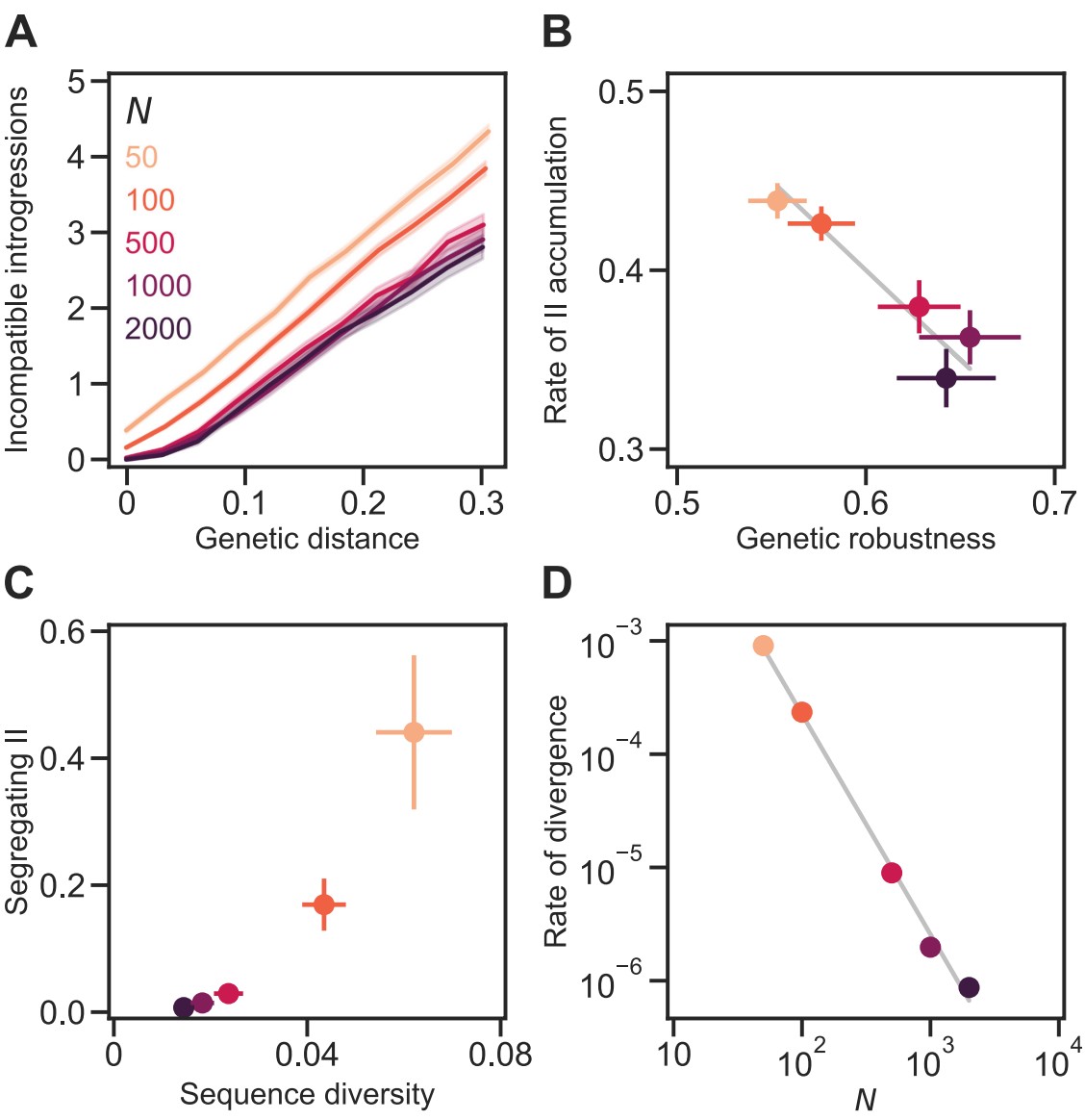

**Fig 5. Genetic drift promotes the accumulation of genetic incompatibilities independently of mutational supply. (A)** IIs accumulated linearly with $D$ in sexual populations of different sizes ($N$). In all simulations, populations evolved under the Russian roulette model (with $p = 0.5$ and $L = 30$) and experienced random mating and free recombination. $U$ was changed such that $NU = 10$. **(B)** IIs accumulated faster in smaller populations because they evolved lower $v$. The gray line shows $b/L = 1 - v$ (see Eq 1). **(C)** Smaller populations accumulated more segregating IIs and sequence diversity (defined as the within-population heterozygosity [42]). **(D)** Large populations diverged more slowly. The gray line shows a power law with exponent −1.9. Plotted values in all panels are means of 200 replicate simulation runs at each $N$. Error bands in (A) and bars in (B) and (C) are 95% CIs (in (D) they are hidden by the points). See Fig 4 for more details.

## Genetic robustness

The mutational robustness of a genotype was measured as the proportion $v$ of its single mutations that did not render the genotype inviable (Fig 1C). The robustness of each replicate population was estimated as the average $v$ of 100 individuals sampled at random without replacement (or $N$ individuals when $N < 100$).

### Reproductive isolation

The level of postzygotic reproductive isolation (RI) between two populations is defined as:

$$\mathrm{RI} = 1 - \frac{1}{N}\sum_{i=1}^{N} w_i \quad ,$$

where $w_i$ is the fitness of a genotype resulting from recombination between the $i$th genotypes of each population.

### Phylogenetic analyses

Phylogenetic signal was tested using the R packages phytools 1.0–3 [34] and geiger 2.0.10 [35]. Phylogenetic generalized least-squares regression analyses were done using the R package caper 1.0.1 [36].

### Software

The software used to run all simulations and to analyze the resulting data was written in Python 3.9 with NumPy 1.21.0 [37], SciPy 1.11.3 [38], scikit-learn 1.3.0 [39] and R 4.1.0 [40].

## Results

### Genetic drift promotes the accumulation of genetic incompatibilities

We allowed sexual populations of different sizes—between 10 and 1000 individuals—to evolve under the Russian roulette model with $p = 0.5$, $U = 0.1$, and free recombination until they reached an approximate equilibrium in genetic robustness and sequence diversity. We then made a copy of each resulting population and allowed each pair of populations to diverge in allopatry. Both populations in each pair experienced the same selection regime (see Materials and methods)—a mutation-order speciation scenario [41]. Fig 4 shows the pattern of accumulation of IIs following the population split. Increasing population size caused the accumulation of IIs to slow down (Fig 4A). Fitting the linear regression model in Eq 3 to the data at each population size revealed a reduction of the slope $b$ with population size (Fig 4B). For example, populations of $N = 500$ accumulated IIs at $\sim 75\%$ of the rate of populations of $N = 10$ individuals.

In larger populations genetic drift is expected to be weaker allowing selection for genetic robustness to operate more efficiently (Fig 3: $N \rightarrow \nu$). Indeed, larger populations evolved higher genetic robustness, $\nu$, before populations began diverging from each other (Fig 4B). The level of genetic robustness in diverging populations explained the variation in the rate of II accumulation in populations of different sizes: $b/L \approx 1 - \nu$ (see Eq 1; Fig 4B, gray line).

Population size had another effect on the rate of speciation: larger populations diverged more slowly from each other (Fig 4D). The rate of increase in $D$ per generation was approximately proportional to $1/N^2$. Populations of $N = 1000$ individuals diverged even more slowly and plateaued at a genetic distance of $D = 0.07 \pm 0.016$ (mean and 95% CI based on 100 simulations over $2 \times 10^5$ generations). In the large population limit ($N \rightarrow \infty$) both populations will be exactly identical and will never diverge ($D = 0$). As a result, any IIs between populations will also be IIs segregating within populations. Both effects of $N$ on the rate of speciation act in the same direction.

The results summarized in Fig 4 do not disentangle the effects of genetic drift, $N$, and mutational supply, $NU$, because the mutation rate was constant ($U = 0.1$). Furthermore, high $U$ is also expected to promote the evolution of high genetic robustness [22, 43] (Fig 3: $U \rightarrow \nu$). Fig 5 shows the results of similar evolutionary simulations where $N$ was changed while keeping a

constant $NU = 10$. The results show that $N$ has an effect on the evolution of genetic robustness independent of mutational supply: increasing $N$ while reducing $U$ selects for higher $v$, which leads to slower accumulation of IIs (Fig 5B). This effect appears to saturate when $N \approx 1000$. We conclude that genetic drift promotes the accumulation of genetic incompatibilities.

## The RNA folding model

One limitation of the Russian roulette model is that it is unrealistic in at least two ways: real fitness landscapes are not completely random and viable genotypes can display intrinsic differences in fitness. To address this limitation we have investigated evolution under a more complex genotype-phenotype map, in which genotypes are RNA sequences and phenotypes are the minimum free-energy secondary structures of those sequences. The fitness of a genotype varies quantitatively with the difference between its phenotype and an optimal phenotype (Eq 2; Fig 6, inset). Genotypes with phenotypes that differ too much from the optimum are inviable. Populations evolving under this model evolve realistic distributions of mutational effects (DFEs). For example, Fig 6 (main) shows the DFE of sexual populations of $N = 100$ individuals after 1,000 generations of evolution with a mutation rate of $U = 0.1$ and free recombination. The mean effect of a nonlethal mutation was $\bar{s} = -0.051$. This is consistent with

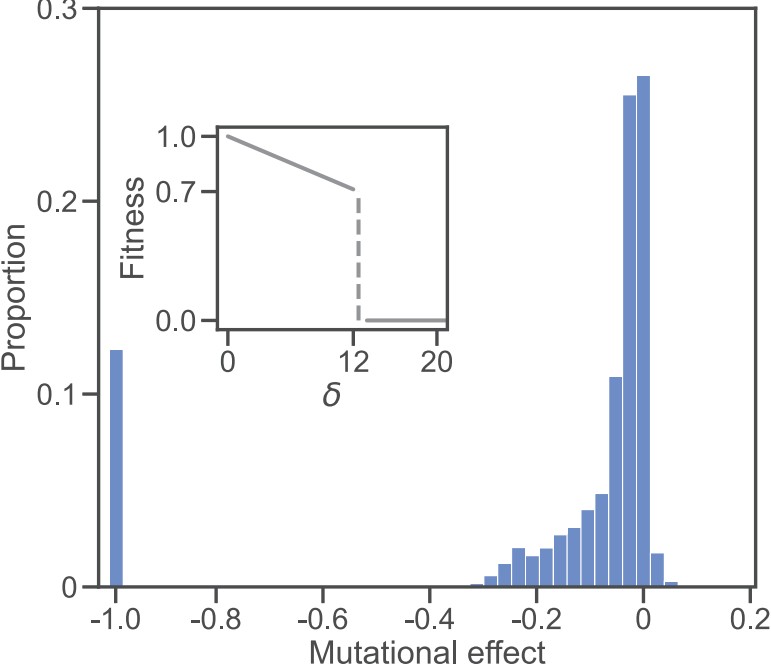

**Fig 6. Distribution of fitness effects (DFE) in the RNA folding model. (Inset)** Fitness landscape. Genotypes are RNA sequences of length $L = 100$. The fitness of a sequence is defined by the base-pair distance $\delta$ between its minimum free-energy secondary structure and a reference structure (Eq 2). A perfect structural match ($\delta = 0$) yields the maximum fitness ($w = 1$). Fitness declines linearly with $\delta$ until a threshold is reached ($\delta = 12$). Larger structural distances ($\delta > 12$) are lethal ($w = 0$). **(Main)** The histogram shows the DFE based on the effects of $10^6$ individual single nucleotide substitutions. One hundred replicate populations of $N = 100$ individuals were allowed to evolve for 1,000 generations under $U = 0.1$, random mating, and free recombination. Each individual in each population was then subject to 100 random mutations. Mutational effects are defined as $s = w'/w - 1$, where $w'$ and $w$ are the fitnesses of the mutant and unmutated genotypes, respectively. The majority of mutations (71.1%) were deleterious ($s < 0$); 12.3% were lethal ($s = -1$); 26.5% were neutral ($s = 0$); 2.4% were beneficial ($s > 0$). The largest beneficial effect was $s = 0.429$ but fewer than 0.1% of mutations had effects $s > 0.2$. The mean effect of a mutation was $\bar{s} = -0.168$; excluding lethals, it was $\bar{s} = -0.051$.

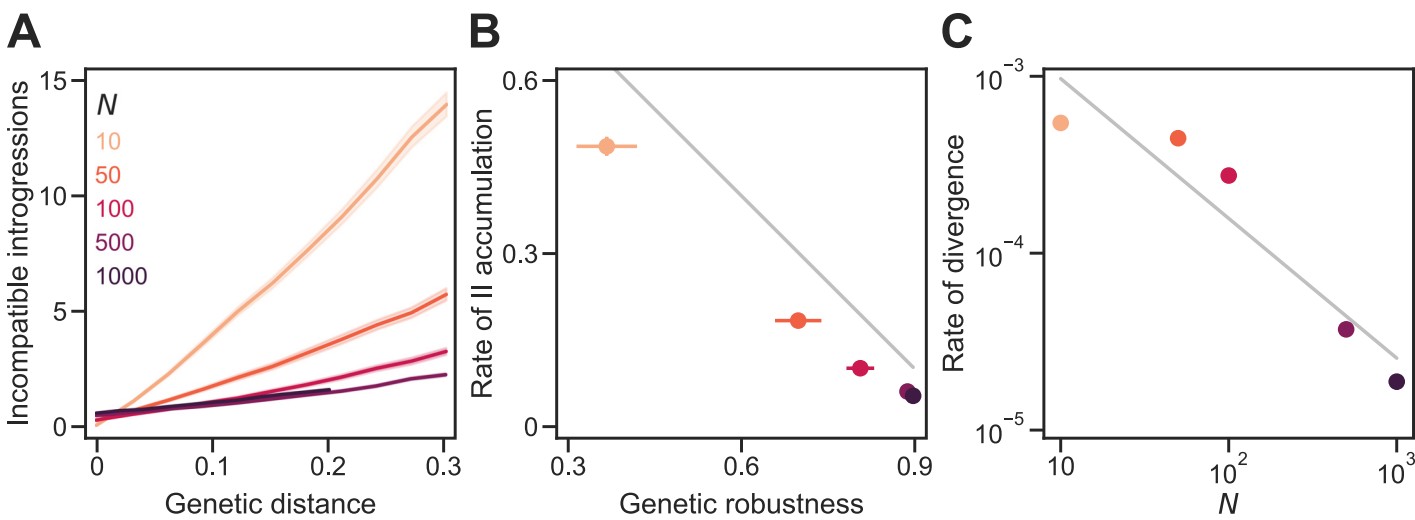

**Fig 7. Genetic drift promotes the accumulation of genetic incompatibilities in the RNA folding model.** (A) IIs accumulated approximately linearly with $D$ in sexual populations of different sizes ($N$). In all simulations, populations evolved under the RNA folding model (with $\sigma = 0.025$ and $L = 100$) and experienced $U = 0.1$, random mating, and free recombination. (B) IIs accumulated faster in smaller populations because they evolved lower $v$. The gray line shows $b/L = 1 - v$ (see Eq 1). (C) Large populations diverged more slowly. The gray line shows a power law with exponent −0.8. Plotted values in all panels are means of 200 replicate simulation runs at each $N$. Error bands in (A) and bars in (B) are 95% CIs (in (C) they are hidden by the points). See Fig 4 for more details. See S1 Fig for the results of simulations on the RNA model with $\sigma = 0$.

estimates from mutation accumulation experiments in a wide variety of organisms [44]. For example, the average mutational effect in 6 species of bacteriophage was $\bar{\bar{s}} = -0.049$ [45]. In the rest of the paper we investigate this model.

In Fig 7 we repeat the analysis in Fig 4 but with the RNA model. Broadly, the results are consistent with those from the Russian roulette model with two important differences. First, $N$ has a stronger effect on the evolution of genetic robustness in the RNA model (Fig 7B). Genetic robustness at equilibrium was 240% higher at $N = 500$ than at $N = 10$ in the RNA model, compared to only 23% higher in the Russian roulette model. These differences in $v$ cause similarly dramatic differences in the rate of II accumulation (a 780% increase in $b/L$ in the RNA model compared to a 33% increase in the Russian roulette model with the change from $N = 500$ to $N = 10$; Fig 7B). Second, the rate of II accumulation ($b/L$) in the RNA model is lower than the expectation from Eq 1 of $1 - v$. The likely reason is that mutational effects are correlated in similar genotypes in the RNA model, unlike the Russian roulette model. Thus, derived alleles that are viable in the genetic background in which they originated are more likely to also be viable in similar genetic backgrounds independently of genetic robustness.

We tested that the genetic incompatibilities that evolved in the RNA model generated postzygotic reproductive isolation (RI) between them by measuring the fitness of hybrids between diverged populations. We found that populations evolved at different population sizes evolved RI proportional to the number of IIs between them. For the data shown in Fig 7 at the end of divergence, adding an II between populations caused an increase in postzygotic RI of approximately 4.5%. The results from the RNA model confirm that genetic drift promotes speciation.

### Recombination hinders the accumulation of genetic incompatibilities

Changing the recombination probability in populations of the same size ($N = 100$) allows us to isolate the causal role of recombination in determining the rate of II accumulation. Populations experiencing stronger recombination both evolved higher genetic robustness and

accumulated IIs more slowly (Fig 3: $r \rightarrow v$; Fig 8A and 8B). For example, populations in which all matings experienced one crossover ($r = 1$) evolved 24% higher robustness and accumulated IIs at 55% the rate of populations in which only 1% of matings experienced one crossover ($r = 0.01$). Increasing the *amount* of recombination from one crossover to free-recombination caused a further increase in robustness of 5% and a further reduction in the rate of II accumulation of 18%.

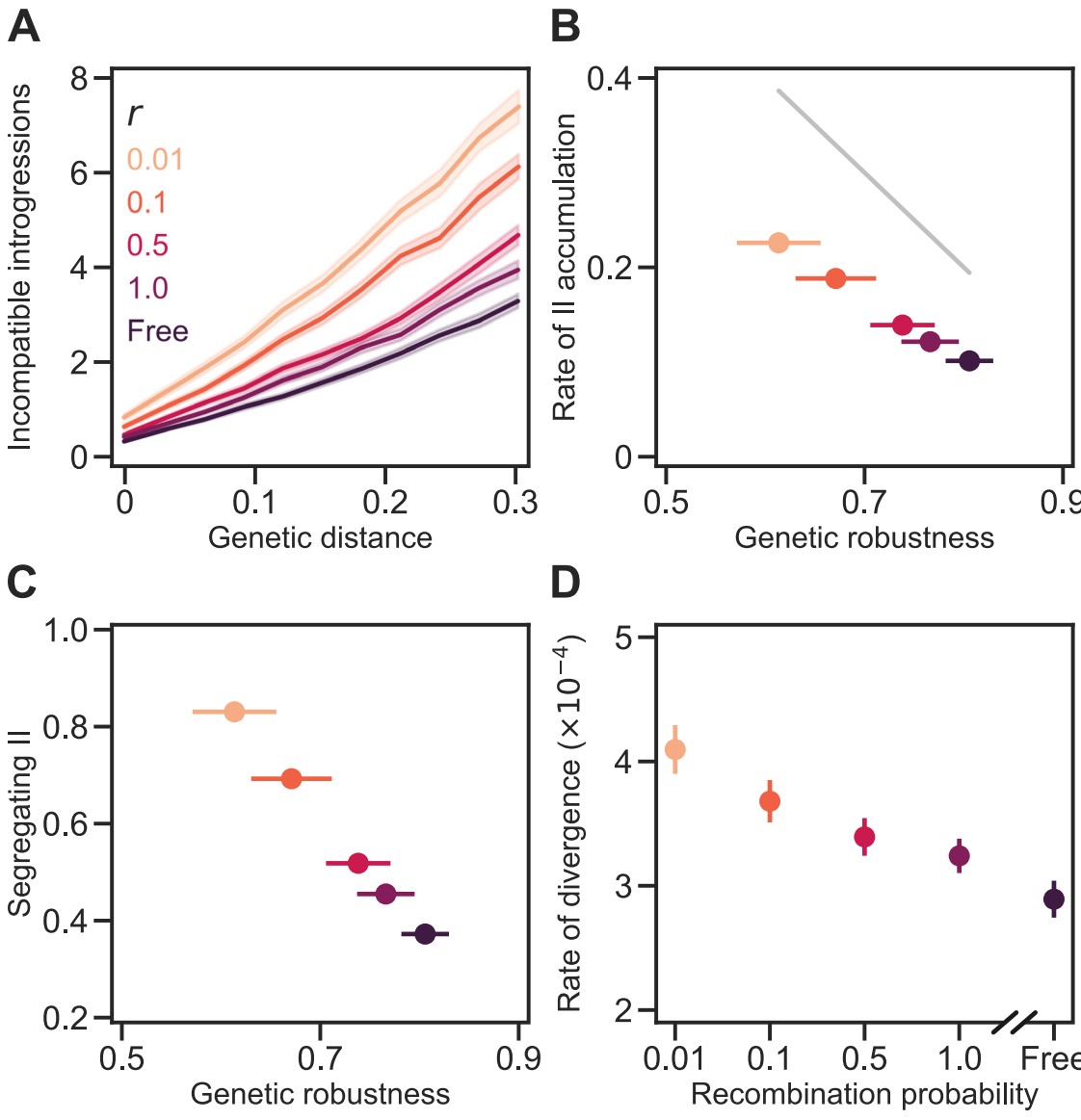

**Fig 8. Recombination hinders the accumulation of genetic incompatibilities. (A)** IIs accumulated approximately linearly with $D$ in populations of $N = 100$ individuals experiencing different recombination probabilities. In all simulations, populations evolved under the RNA folding model (with $\sigma = 0.025$ and $L = 100$) and experienced $U = 0.1$ and random mating. The values of the recombination probability ($r = 0.01, \ldots 1$) indicate the expected proportion of matings resulting in a recombinant progeny containing one crossover event. Under free recombination, every progeny is a recombinant ($r = 1$) with an expected 49.5 crossover events. **(B)** IIs accumulated faster in populations experiencing low recombination probability because they evolved lower $v$. The gray line shows $b/L = 1 - v$ (see Eq 1). **(C)** Populations experiencing low recombination probability accumulated more segregating IIs. **(D)** Populations experiencing high $r$ diverged more slowly. Values show the rate of increase in $D$ per generation. Plotted values in all panels are means of 200 replicate simulation runs at each $r$. Error bands in (A) and bars in (B)–(D) are 95% CIs (some are hidden by the points). See Fig 4 for more details. See S2 Fig for the results of simulations on the RNA model with $\sigma = 0$.

The previous results indicate that recombination hinders the accumulation of genetic incompatibilities. But how does recombination do that? To begin to answer that question we consider the accumulation of IIs in the absence of recombination. In sufficiently large populations, genetic variation builds up. Because of epistatic interactions, some of these variants constitute within-population IIs (Fig 3: $\pi \to II_W$). In the absence of recombination, however, these segregating genetic incompatibilities will only rarely be expressed—and, therefore, be exposed to natural selection—because they can only occur in the same individual through multiple mutations, which are rare. Therefore, cryptic IIs are expected to accumulate *within* asexual populations.

We hypothesize that the extent to which a mutation is involved in an II *within* a population is an important determinant of whether that mutation will cause an II *between* populations if it spreads in the population (Fig 3: $II_W \to II_B$). To test this hypothesis we isolated from each asexual population ($r = 0$) the first mutation to rise to high frequency and cause an II among populations and the first mutation to rise to high frequency and not cause an II. We then went back to the generations in which these mutations arose, placed them in every individual present in the population at the time, and noted the proportion of these "backcrossed" individuals that remained viable (Fig 9A). As predicted, we found that the proportion of viable backcrossed individuals was much lower for mutations that caused IIs (0.75±0.03, mean and 95% CI; Fig 9B, $r = 0$ filled circle) than for mutations that did not (0.91±0.02; Fig 9B, $r = 0$ open circle). In

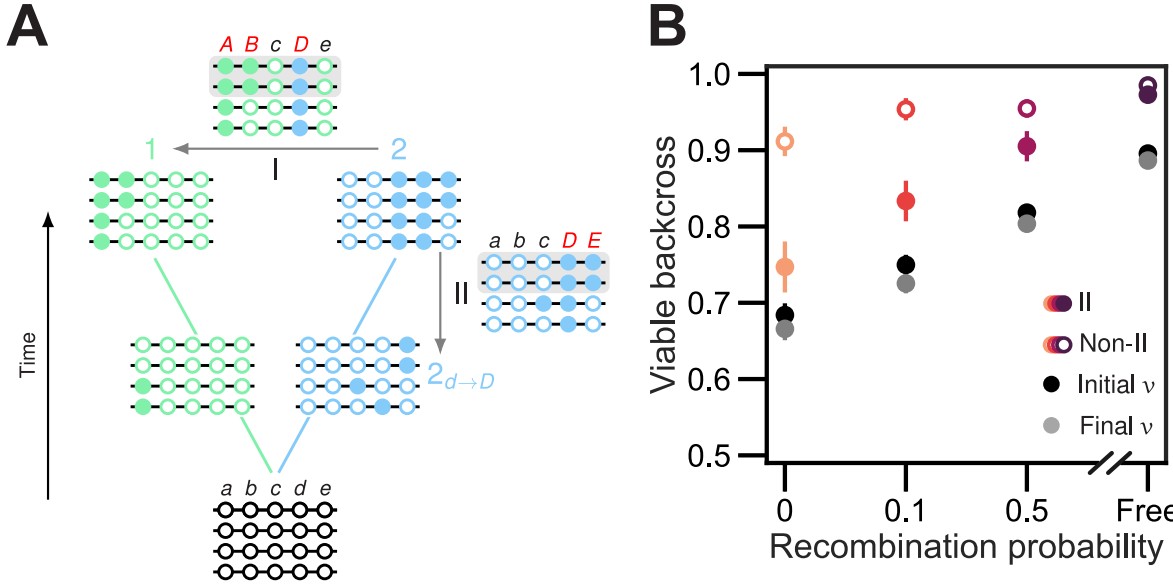

**Fig 9. Recombination selects for alleles that are more compatible with the genetic backgrounds of other individuals in the population.** (A) Illustration of the experimental design used to generate data in (B). Two populations of $N = 4$ individuals diverge in allopatry over the holey fitness landscape shown in Fig 1C. After some time, population 2 (blue) fixes derived allele *D*. Introgressing *D* from population 2 to population 1 (arrow I) results in an inviable genotype (*ABcDe*), rendering 50% of introgressed sequences in population 1 inviable (gray box). Backcrossing *D* into every genotype present in population 2 (arrow II) when the *D* allele arose by mutation reveals an inviable genotype (*abcDE*), and renders 50% of the sequences in that population inviable too (gray box). Thus, *D* is involved in IIs both within and among populations. (B) We repeated simulations like those summarized in Fig 8 (but including an asexual treatment, $r = 0$) and isolated the first mutation to occur in a population, rise to a frequency of at least 50%, and be involved in an II with the other population. We then went back to the generation in which the mutation arose and backcrossed it into every individual present within its own population at that time. Panel (B) shows the proportion of viable backcrossed individuals made from populations experiencing different recombination probabilities (colored filled circles). As a control, we did the same calculation for the first mutation to rise to high frequency that does not cause an II among populations (open circles). Black and gray circles show the mean genetic robustness of the populations when the mutations arise and when they reach high frequency, respectively. Values are means of 200 replicate simulation runs at each recombination probability. Error bars are 95% CIs.

other words, mutations that caused IIs among asexual populations were likely to have started out causing cryptic IIs within populations.

We propose that recombination slows down the accumulation of IIs *between* populations because it suppresses the accumulation of IIs *within* populations. We found that sexual populations with more genetic variation within populations displayed greater numbers of segregating IIs (Fig 5C). In the presence of recombination, segregating IIs are not cryptic; they are expressed in recombinant individuals. This suppresses the accumulation of IIs within populations in two ways. First, it causes selection against mutations involved in IIs within populations (Fig 3: $r \dashv II_W$). Second, it generates recombination load which, in turn, selects for genetic robustness [19, 46–50]. Genetically robust individuals are less likely to express IIs (Fig 3: $r \rightarrow v \dashv II_W$). We observe this suppression of segregating IIs by recombination in our simulations (Fig 8C) and believe the reduced supply of segregating IIs slows down the accumulation of IIs among populations (Fig 3: $II_W \rightarrow II_B$).

We tested this hypothesis by conducting backcross simulations at different recombination rates (Fig 9B). As recombination rate and robustness increased, we observed a higher proportion of viable backcrossed individuals for mutations that caused IIs among populations (Fig 9B, orange filled circles). Thus, in the presence of recombination, even mutations that are ultimately destined to participate in genetic incompatibilities among populations are less likely to result in within-population genetic incompatibilities (Fig 3). That is because recombination selects for genotypes that are more compatible with the genetic backgrounds of other individuals in the population and, therefore, with mutations currently segregating in the population.

### *Drosophila* species with higher levels of neutral polymorphism and higher recombination rate accumulate postzygotic RI more slowly

Our results lead to the prediction that the rate with which postzygotic RI accumulates with genetic distance—RI velocity [51]—should be negatively correlated with both population size and recombination rate. We tested these predictions using published estimates of postzygotic RI velocity in *Drosophila* species groups [51]. We estimated the level of neutral polymorphism at the *Adh* locus—a proxy for effective population size—within 38 species belonging to 8 of the species groups and found a significant negative correlation between RI velocity and the average level of neutral polymorphism in a species group (linear regression: $R^2 = 0.71$, $P = 0.008$; Fig 10B). Correcting for phylogenetic autocorrelation only strengthened the effect ($P = 0.002$).

We also estimated the genome-wide recombination rate of 15 species in the same 8 species groups and found a negative correlation between postzygotic RI velocity and the average recombination rate in a species group, after removing the effect of the level of neutral polymorphism (Fig 10C). Including recombination rate in the linear model increased the proportion of variance in RI velocity explained from $R^2 = 0.71$ to $0.84$. However, this improvement was not statistically significant ($P = 0.11$).

### Differences in the level of neutral polymorphism of two species explain asymmetry in postzygotic RI between them

Another prediction from our work is that differences in the size of diverging populations will cause asymmetric accumulation of genetic incompatibilities: the expected number of IIs will be greater in introgressions from a large population to a small population than in the opposite direction. Asymmetric postzygotic RI has been observed between reciprocal crosses of many closely related species—a phenomenon termed Darwin's corollary to Haldane's rule [56]. We propose that this kind of asymmetry might be explained by differences in robustness to introgression between the recipient populations. Yukilevich [53] surveyed RI asymmetry between

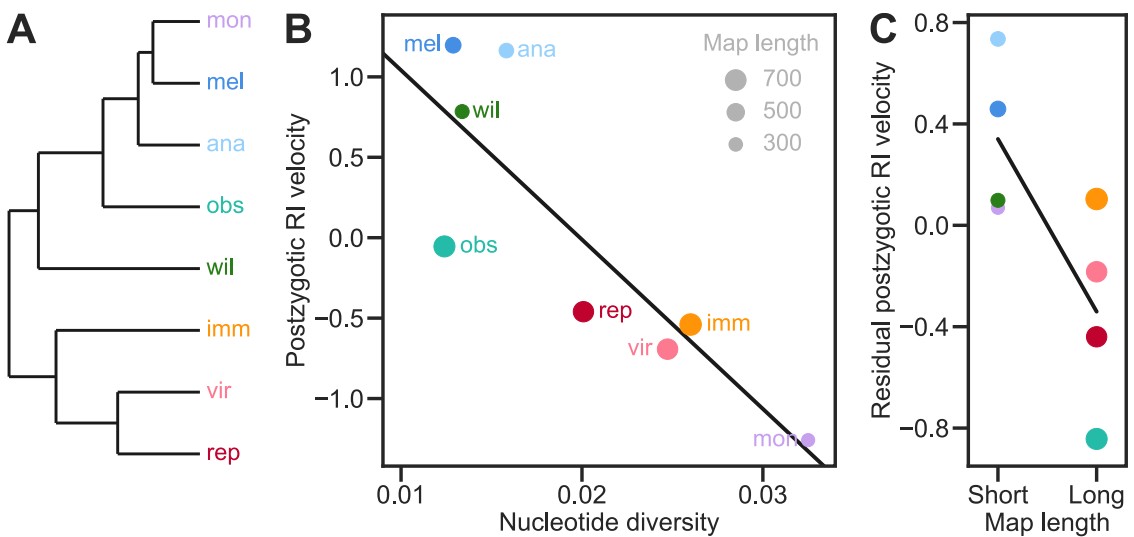

**Fig 10. *Drosophila* species with higher levels of neutral polymorphism and higher recombination rate accumulate postzygotic RI more slowly.** (A) Phylogenetic relationships between 8 *Drosophila* species groups [51] (*ananassae*, *immigrans*, *melanogaster*, *montium*, *obscura*, *repleta*, *virilis*, *willistoni*). (B) The rates with which postzygotic RI accumulates with genetic distance in the species groups shown in (A) were estimated by Rabosky and Matute [51] based on the proportion of sterile or inviable $F_1$ offspring in interspecific crosses [52, 53]. Positive and negative values represent faster- and slower-than-average RI velocities. The levels of neutral polymorphism in individual species were measured by the synonymous nucleotide diversity ($\pi_s$) at the *Adh* locus (see S1 Table). Values are the mean $\pi_s$ for the species or subspecies in each *Drosophila* species group. Multiple estimates for a single species were averaged before calculating the overall mean for the species group. The line shows a fit by linear ordinary least-squares regression of RI velocity against $\pi_s$ (OLS: slope and 95% CI = −105±67, $R^2$ = 0.71, P = 0.008). The results of a phylogenetic generalized least-squares regression analysis assuming Brownian motion were almost identical (PGLS: $R^2$ = 0.82, P = 0.002). (C) Residuals in RI velocity from the regression in (B) in the species groups with either shorter or longer total map lengths than average. The point areas in (B) and (C) are proportional to the mean total map length in cM in the species group (see S2 Table). Adding map length to the regression model did not improve fit in a statistically significant manner (multiple regression OLS: coefficient and 95% CI = −0.0016±0.0021, $R^2$ = 0.84, P = 0.11; PGLS: $R^2$ = 0.89, P = 0.13). None of the traits showed a significant phylogenetic signal based on either the $K$ or $\lambda$ statistics [54, 55] (see S3 Table).

many *Drosophila* species. We were able to determine the level of neutral polymorphism within both species of 8 species pairs in which he found asymmetric postzygotic RI. The pattern of asymmetry in RI between these species pairs showed a statistically significant trend in the direction we predicted (linear regression through the origin: $R^2$ = 0.52, P = 0.028; Fig 11).

## Discussion

Rosenzweig [57] noted that "no one denies that population size influences speciation rate. However, the direction of its effect is in doubt." In his classic neutral theory of biodiversity, Hubbell [58] assumed that all species have the same speciation rate per capita and that, therefore, more abundant species will have higher speciation rates. Similarly, Marzluff and Dial [59] proposed that a broad geographic distribution—a correlate of population size—promotes speciation. In contrast, Stanley [60] hypothesized that species whose populations contract and fragment due to, for example, predation or environmental deterioration, tend to experience high rates of speciation.

Some attempts to study this issue using explicit theoretical models have reached the same conclusion as ours, that low abundance promotes speciation. However, the models they considered differ from ours in important aspects. Mayr [61] proposed that speciation often results from "genetic revolutions" precipitated by founder events. Others have developed similar

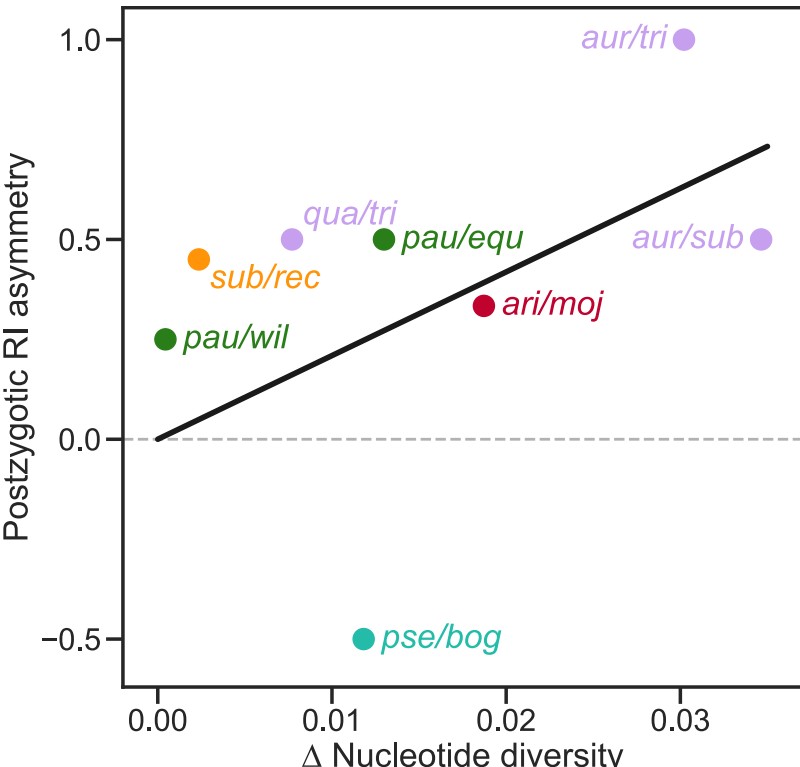

**Fig 11. Differences in the level of neutral polymorphism of two species explain asymmetry in postzygotic RI between them.** Postzygotic RI asymmetry for the $sp1/sp2$ pair was estimated [53] by taking the RI score from a ♀ $sp1$ × ♂ $sp2$ cross and subtracting the RI score from the reciprocal cross. The x-axis shows the difference in the synonymous nucleotide diversity ($\pi_{s1} - \pi_{s2}$) at the *Adh* locus (except the *rec/sub* pair for which we used the *Adhr* locus; see S4 Table). The colors of the points match the species groups shown in Fig 10A except that the *quinaria* group (sister group to *immigrans*) is represented by orange: *montium* group: <u>*auraria*</u>, <u>*quadraria*</u>, <u>*subauraria*</u>, <u>*triauraria*</u>; *obscura* group: <u>*bogotana*</u>, <u>*pseudoobscura*</u>; *quinaria* group: <u>*recens*</u>, <u>*subquinaria*</u>; *repleta* group: <u>*arizonae*</u>, <u>*mojavensis*</u>; *willistoni* group: <u>*equinoxialis*</u>, <u>*paulistorum*</u>, <u>*willistoni*</u>. The line shows a fit by linear ordinary least-squares regression through the origin (slope and 95% CI = 20.9±17.8, $R^2$ = 0.52, $P$ = 0.028).

models [62]. Most of them modeled ecological speciation and did not consider genetic incompatibilities as the basis for RI. Gavrilets and Hastings [63] did consider several DMI scenarios and showed that founder effect speciation is "plausible" under some of them, but did not consider the accumulation of multiple DMIs of arbitrary complexity.

Maya-Lastra and Eaton [64] modelled genetically variable populations evolving in the presence of multiple pairwise DMIs like those envisioned by Orr and, like us, observed that smaller populations accumulated more DMIs. They incorporated recombination in their model but did not investigate its effect. They assumed that mutations involved in DMIs were beneficial but that the DMIs were weak, unlike in our models.

Khatri and Goldstein showed that genetic incompatibilities accumulated more quickly in smaller populations in several models of gene regulation [65–67]. Like our models, their models included strong incompatibilities that create holes in the fitness landscape. Furthermore, the fitness of a viable genotype is a continuous function of the phenotype in their models, like in our RNA folding model. Khatri and Goldstein have argued that their results arise because smaller populations evolve a greater drift load and occupy regions of genotype space closer to the holes in the fitness landscape. We believe that the same mechanism operates in our RNA

folding model. However, that is not the case in the Russian roulette model, indicating that additional mechanisms may be at play. One limitation of the Khatri and Goldstein studies is that they considered only the Weak Mutation regime and, therefore, were not able to evaluate the effect of recombination. Furthermore, under Weak Mutation, neither genetic drift nor recombination would have any effect on the rate of accumulation of incompatibilities in our models. Therefore, our results must also involve evolutionary mechanisms not captured in the models studied by Khatri and Goldstein. We believe the ingredient they missed is segregating variation at loci involved in genetic incompatibilities. Such variation has been known to exist in natural populations of several species [68, 69]. Corbett-Detig et al. [69] found evidence that multiple pairwise DMIs are currently segregating within natural populations of *D. melanogaster*. They surveyed a large panel of recombinant inbred lines (RILs) and found 22 incompatible pairs of alleles at unlinked loci in the RILs; of the 44 alleles, 27 were shared by two or more RILs, indicating that multiple DMIs are polymorphic within natural populations. They also found evidence for multiple DMIs in RIL panels in *Arabidopsis* and maize [69]. Corbett-Detig et al. [69] did not attempt to identify DMIs among linked loci or higher-order DMIs and therefore are likely to have underestimated the actual number of segregating DMIs in the RILs. These observations suggest that the conditions for the effects reported here may occur in nature.

Not all previous modeling studies agreed with ours. Gavrilets et al. [7] studied the effect of population size on the rate of speciation in a holey fitness landscape model using simulations. Individuals differing at a threshold number, $K$, of diallelic loci were considered incapable of producing viable offspring. They found that when selection was strong (low $K$) small population size promoted speciation; in contrast, when selection was weak (high $K$) large population size promoted speciation. Their model differs from ours in two important ways. First, it does not distinguish between pre- and postzygotic RI [7]. Two genotypes differing at more than $K$ loci might simply be incapable of mating with each other. Second, it does not consider the possibility that pairs of genotypes at a certain genetic distance may display different levels of postzygotic RI, a crucial feature of our models.

Another contrary result was obtained by Orr and Orr [70]. They modeled a population of $N$ individuals divided into $d$ geographically isolated subpopulations of equal size ($N/d$). They assumed: (1) that the subpopulations evolve according to Orr's [28] combinatorial model (2) in the Weak Mutation regime; (3) that all DMIs occur between pairs of alleles; (4) that there is a fixed probability, $q$, that a new substitution in one subpopulation is incompatible with each of the $k-1$ divergent alleles in another subpopulation; and (5) that two subpopulations become different species once the cumulative number of DMIs between them reaches some threshold. Orr and Orr [70] calculated the expected number of substitutions, $\hat{k}$ required for speciation to occur. They found that, when the DMIs were caused by neutral substitutions, subpopulation size had no effect on $\hat{k}$. In contrast, when the DMIs were caused by beneficial substitutions, $\hat{k}$ was inversely related to subpopulation size; in other words, larger subpopulation size *promoted* speciation. Both of these results contradict the findings of our paper. We believe that the discrepancy is explained by the fact that Orr and Orr [70] assumed both Weak Mutation and that the DMI probability $q$ was constant over different levels of population subdivision. These assumptions preclude the possibility that $q$ might evolve because of changes in genetic robustness as in our models.

The results from our study lead to a clear prediction that postzygotic RI should build up more quickly in smaller populations. There have been multiple attempts to test this prediction experimentally by subjecting populations to founder events, sometimes repeatedly. In one of these studies, Ringo et al. [71] subjected 8 experimental populations of *D. simulans* to repeated

bottlenecks of a single pair of individuals every 6 months. Between bottlenecks these "drift" lines were allowed to expand to large size. Another 8 populations were maintained at a larger size and subjected to a variety of artificial selection regimes. The drift lines experienced stronger inbreeding than the "selection" lines. An ancestral "base" population was kept at larger size than either treatment group throughout. Ringo et al. found that after 6 bottleneck/expansion cycles the drift lines evolved stronger postzygotic RI relative to the base population than did the selection lines (28% vs 4% reduction in hybrid fitness), in agreement with our prediction. However, the results from three similar experiments on other species of *Drosophila* were inconclusive [72–75].

There is also comparative evidence supporting our prediction. Huang et al. [76] tested for correlations between a measure of the level of neutral polymorphism within species—a proxy for effective population size, $N_e$—and speciation rates estimated from phylogenies in three genera of lichen-forming fungi. They found a significantly negative correlation in one genus, in agreement with our prediction, but no significant correlations in the other two genera. A larger phylogenetic study found significant negative correlations between speciation rates and a different proxy for $N_e$—geographic range size—in both birds and mammals [64]. These results, while encouraging, have the limitation that they have used overall speciation rate, which may not be driven primarily by postzygotic RI in all lineages. Furthermore, the estimates of speciation rate may themselves be problematic because many different temporal patterns of speciation and extinction rates can have equal probability for a given phylogeny [77, 78]. To address these limitations we tested for a correlation between the level of neutral polymorphism within species ($\pi_s$) and the rate with which postzygotic RI accumulates with genetic distance in *Drosophila* [51]. We found a negative correlation (Fig 10B). One potential limitation of this analysis is that we averaged $\pi_s$ across species within a *Drosophila* species group and some species groups (e.g., the *melanogaster* group) show considerable variation in $\pi_s$. We also found that differences in the level of neutral polymorphism of two species explained asymmetry in postzygotic reproductive isolation between them (Fig 11). Note, however, that these comparative correlations could have alternative explanations. For example, reductions in population size might be a *consequence* of speciation, rather than its cause [79].

As with population size, no one denies that recombination influences speciation rate. But, again, the direction of its effect is disputed. Butlin [80] argued that "speciation can be viewed as the evolution of restrictions on the freedom of recombination". In general, low recombination is found to promote speciation in the *presence* of gene flow between diverging populations because it restricts introgression. For example, chromosomal rearrangements often suppress recombination along a large region of the genome. Navarro and Barton [81] showed that such chromosomal rearrangements facilitate the accumulation of DMIs in the presence of gene flow. Interestingly, genetic drift can help maintain genetic differentiation within inversions between populations coming into secondary contact [82]. However, recombination does not always hinder speciation. Bank et al. studied the effect of recombination on the maintenance of a DMI involving two loci in a haploid, with gene flow from a continental to an island population [83]. They showed that recombination can have opposing effects on the maintenance of the DMI depending on the fitnesses of the genotypes at the DMI loci. In some scenarios, the DMI was maintained more easily through selection against immigrants when recombination was low. But in other scenarios, a DMI was maintained more easily through selection against hybrids when recombination was high. Founder effect speciation models have also tended to conclude that high recombination should promote speciation [84]. In contrast to the results from earlier models, we found that recombination within populations can hinder speciation in the *absence* of gene flow. Our results indicate that recombination suppresses the accumulation

of genetic incompatibilities within populations, which in turn slows down their accumulation between populations.

Our results lead to the prediction that recombination in large populations impedes speciation; such populations would maintain extensive genetic variation but even highly divergent genotypes would remain compatible with each other. This prediction is consistent with the observation that recombination between highly divergent genotypes is common in viruses, including influenza A [85, 86], HIV-1 [87, 88], hepatitis B [89] and hepatitis E [90] viruses, rhinoviruses [91, 92], coronaviruses [93–99], and geminiviruses [100]. Such recombination events have often contributed to the emergence of viral diseases. For example, the virus that caused the 1918 influenza pandemic is believed to have arisen when a human virus containing an H1 hemagglutinin acquired an N1 neuraminidase from an avian virus [86].

We also obtained limited comparative evidence for the prediction that recombination slows down the accumulation of postzygotic RI. We found a negative correlation (not statistically significant) between the genome-wide recombination rate and the rate with which postzygotic RI accumulates with genetic distance in *Drosophila* species groups [51], after removing the effect of the level of neutral polymorphism (Fig 10C).

We found that microevolution and macroevolution were *coupled* in our models. This coupling is also a feature of our result that genetic drift promotes speciation. In both cases, processes acting within populations to alter genetic robustness explain variation in speciation rates, contrary to Stanley's [2] contention that speciation is largely random. The size of a population is also predicted to be inversely related to its probability of extinction [101–104]. Thus, if our results are correct, depending on the balance of the effects of population size on speciation and extinction, random genetic drift within species could be a driver of deterministic species selection [3], the process championed by Stanley [2].

We identified a crucial role of genetic robustness in the accumulation of genetic incompatibilities in our models. We showed that high recombination rate selected for genetic robustness in the RNA folding model. This result was obtained previously using simulations on: models of RNA folding [20] and gene networks [19, 48, 50, 105], the Russian roulette model [21], and digital organisms [47]; it was also derived analytically in a modifier model [46]. We also showed that large population size selected for genetic robustness in both the RNA and Russian roulette models. This result is more surprising. Theory and simulations using a quasispecies model [106] and simulations using digital organisms [107] showed that large populations tended to evolve *lower* genetic robustness. In contrast, simulations using a gene network model showed that large populations evolved higher genetic robustness [105]. The difference may have stemmed from a trade-off between fitness and genetic robustness, that was explicit in the quasispecies model [106] and emergent in the digital organism simulations [107], but did not occur in our models or in the gene network model [105]. The extent to which genetic robustness carries costs in real organisms remains an open question.

Environmental stochasticity selects directly for environmental robustness and indirectly for genetic robustness [22, 24]. Therefore, we predict that temporal variation in environmental conditions should hinder speciation. This prediction is consistent with the observation that speciation rates are higher in tropical lineages than in temperate lineages of amphibians [108], birds [109], and mammals [110]. Furthermore, Yukilevich [111] found that postzygotic RI (but not prezygotic RI) accumulated more than twice as quickly between tropical species pairs of *Drosophila* than between temperate and subtropical species pairs of *Drosophila*. However, many other hypotheses have been proposed to explain these patterns [112, 113].

A striking feature of our results is that the simulations in the RNA folding model resulted in approximately linear patterns of II accumulation. Fitting a nonlinear model of the form $II = a + b(LD − 1)^c$ to the 9 sets of simulations summarized in Figs 7 and 8 yielded values of

the exponent *c* between 1.19 and 1.47. An exploration of the RNA folding model without intrinsic fitness differences ($\sigma = 0$) in the Weak Mutation regime resulted in an exponent of *c* = 1.35 [31]. These values of *c* are mostly higher than those predicted under the Russian roulette model (*c* = 1). This is not surprising because the fitness landscape in the RNA folding model is not random. However, the observed values of *c* are also lower than predicted by the combinatorial model proposed by Orr [28] according to which the exponent should be *c* ≳ 2—a pattern he called "snowballing". We believe that the mismatch is caused by the fact that a central assumption of Orr's model is violated in the RNA folding model: that DMIs, once they have arisen, should persist indefinitely regardless of future evolutionary change [31].

The extent to which our results can be generalized to real organisms requires further investigation. Our models make several strong assumptions, such as, that strong epistatic interactions are common, haploid, and that populations of constant size diverge in allopatry under the same selection regime (i.e., mutation-order speciation [41]). The effect of each of these assumptions can be investigated within the framework of our models. Such an exploration has the potential to serve as the foundation for a new theory of the coupling between microevolution and macroevolution.

## Supporting information

**S1 Fig. Genetic drift promotes the accumulation of genetic incompatibilities in the RNA folding model without intrinsic fitness differences between genotypes. (A)** IIs accumulated approximately linearly with *D* in sexual populations of different sizes (*N*). In all simulations, populations evolved under the RNA folding model (with $\sigma = 0$ and *L* = 100) and experienced *U* = 0.1, random mating, and free recombination. **(B)** IIs accumulated faster in smaller populations because they evolved lower *v*. The gray line shows $b/L = 1 - v$ (see Eq 1). **(C)** Large populations diverged more slowly. The gray line shows a power law with exponent −0.5. Plotted values in all panels are means of 200 replicate simulation runs at each *N*. Error bands in (A) and bars in (B) are 95% CIs (in (C) they are hidden by the points). See Fig 7 for more details.
(PDF)

**S2 Fig. Recombination hinders the accumulation of genetic incompatibilities in the RNA folding model without intrinsic fitness differences between genotypes. (A)** IIs accumulated approximately linearly with *D* in populations of *N* = 100 individuals experiencing different recombination probabilities. In all simulations, populations evolved under the RNA folding model (with $\sigma = 0$ and *L* = 100) and experienced *U* = 0.1 and random mating. **(B)** IIs accumulated faster in populations experiencing low recombination probability because they evolved lower *v*. The gray line shows $b/L = 1 - v$ (see Eq 1). **(C)** Populations experiencing low recombination probability accumulated more segregating IIs. **(D)** Populations experiencing high *r* diverged more slowly. Values show the rate of increase in *D* per generation. Plotted values in all panels are means of 200 replicate simulation runs at each *r*. Error bands in (A) and bars in (B)–(D) are 95% CIs (some are hidden by the points). See Fig 8 for more details.
(PDF)

**S3 Fig. Number of divergent alleles from one population that have not been tested by natural selection in the genetic background of another population. (A)** Example of scenario A. Two populations diverge in allopatry. Both populations are initially fixed for lowercase alleles (open circles) at four loci *(abcd)*. Derived alleles are indicated by uppercase letters (closed

circles). The donor population undergoes four substitutions, fixing the *ABCD* genotype. The recipient population does not undergo any substitutions, retaining the ancestral genotype *abcd*. Arrows indicate introgressions of divergent alleles from the donor population to the recipient population. Three introgressed genotypes have not been tested by natural selection (*aBcd*, *abCd*, and *abcD*; solid arrows) but one has (*Abcd*, dashed arrow). **(B)** Example of scenario B. Each population undergoes two substitutions, the recipient population fixing the *AbCd* genotype and the donor population fixing the *aBcD* genotype. Three introgressed genotypes have not been tested by natural selection (*abCd*, *ABCd*, and *AbCD*; solid arrows) but one has (*Abcd*, dashed arrow).
(PDF)

**S1 Text. Number of divergent alleles from one population that have not been tested by natural selection in the genetic background of another population.**
(PDF)

**S1 Table. Synonymous nucleotide diversity ($\pi_s$) at the *Adh* locus for different species of *Drosophila*.**
(PDF)

**S2 Table. Total recombination map length for different species of *Drosophila*.**
(PDF)

**S3 Table. Phylogenetic signal for each trait analyzed in Fig 10.**
(PDF)

**S4 Table. Asymmetry in synonymous nucleotide diversity ($\pi_s$) at the *Adh* locus and in postzygotic reproductive isolation (RI) for different species of *Drosophila*.**
(PDF)

## Acknowledgments

The authors thank D. Wiernasz, M. Servedio, and O. Martin for helpful discussions. We used the Opuntia and Sabine clusters from the Hewlett Packard Enterprise Data Science Institute at the University of Houston. We thank the advanced support from the Research Computing Data Core at the University of Houston.

## Author Contributions

**Conceptualization:** Ata Kalirad, Christina L. Burch, Ricardo B. R. Azevedo.

**Funding acquisition:** Christina L. Burch, Ricardo B. R. Azevedo.

**Investigation:** Ata Kalirad, Ricardo B. R. Azevedo.

**Methodology:** Ata Kalirad, Christina L. Burch, Ricardo B. R. Azevedo.

**Project administration:** Ricardo B. R. Azevedo.

**Software:** Ata Kalirad, Ricardo B. R. Azevedo.

**Visualization:** Ata Kalirad, Ricardo B. R. Azevedo.

**Writing – original draft:** Ata Kalirad, Christina L. Burch, Ricardo B. R. Azevedo.

**Writing – review & editing:** Ata Kalirad, Christina L. Burch, Ricardo B. R. Azevedo.

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
