## [Decision Letter · Decision Letter 0]

7 May 2023

Dear Dr Azevedo,

Thank you very much for submitting your Research Article entitled 'Genetic drift promotes and recombination hinders speciation on a holey fitness landscape' to PLOS Genetics.

The manuscript was fully evaluated at the editorial level and by independent peer reviewers. Based on the reviews, we will not be able to accept this version of the manuscript, but we would be willing to review a much-revised version. We cannot, of course, promise publication at that time.

As you will see below, all three reviewers provided detailed, thoughtful feedback and found much to like about this manuscript.  They also each identified substantial concerns.  Among these, it will be critical to put the work in better context of work that has come before (including a clarification of "holey landscape" vs "neutral network"), and to test the robustness of your results to a number of assumptions, as suggested in the reviews.  Key points to be addressed from the reviews include:

- disentangling the effects of drift vs. mutational supply by simulating at fixed NU

- addressing whether the results are apparent because the only selection that can occur is for robustness (if there were intrinsic fitness differences, would these swamp selection for robustness?) 

- going further to address how robust the results are to variation in parameter values and the fitness landscape

- recomputing data for Figs 3 and 4, and addressing concerns re: the snowball exponent as suggested by Reviewer 2 

Should you decide to revise the manuscript for further consideration here, your revisions should also address the specific points made by each reviewer. We will also require a detailed list of your responses to the review comments and a description of the changes you have made in the manuscript.

If you decide to revise the manuscript for further consideration at PLOS Genetics, please aim to resubmit within the next 60 days, unless it will take extra time to address the concerns of the reviewers, in which case we would appreciate an expected resubmission date by email to plosgenetics@plos.org.

We are sorry that we cannot be more positive about your manuscript at this stage. Please do not hesitate to contact us if you have any concerns or questions.

Yours sincerely,

Lindi Wahl

Guest Editor

PLOS Genetics

Gregory Barsh

Editor-in-Chief

PLOS Genetics

Reviewer's Responses to Questions

**Comments to the Authors:**

Reviewer #1: This paper uses simulations of computationally folded RNA populations to evaluate how epistatic incompatibilities accumulate between allopatric populations under the same selection pressures. This is a really interesting topic, and to my knowledge the inclusion of mutational robustness as an evolving factor in this study is relatively novel and leads to insightful results. While I have some notes on the framing, my major concerns are with the approach and interpretation of results. I don’t think these concerns are necessarily indicative of serious problems, but give me pause about recommending the paper as written.

The Introduction is brief and skips past most literature connecting micro-and macroevolution. I don’t think it is necessary to get into a big review of the literature, but as written the Introduction first seems to suggest that few have tried to link micro and macro, then focuses entirely on the accumulation of DMIs, which is, of course, one of the ways people have tried to link them. Prior work on holey fitness landscapes and speciation is mentioned only very briefly in the Intro, and I think here you really should give a little more space to what has been done and what still needs to be done in this arena.

On the actual approach, I guess my first question is why we care about DMIs in asexual populations at all? Maybe I’m out of the loop here, but I don’t see much relevance of the DMI concept to asexual species, which, as we see in Figure 3a, accumulate DMIs even without allopatry. I understand that the point is maybe to compare to sexual populations, but that seems adequately done by reducing r down toward zero, as is done in Figure 4. As such, the whole way the Results begin is a little baffling to me, and could I think be made more concise.

Additionally, I am not sure why genetic drift, and not mutational input, is seen as the causal factor leading to greater mutational robustness. It would seem to be very easy to separate these two–simply perform simulations with a fixed value of NU, decreasing U in concert with increasing N. I think this would be an important test of the authors’ interpretations.

The RNA model, as used here, has the obvious strength of generating epistasis, and I’m certainly not going to suggest any major changes to the core model. But, I’m not sure it’s quite the right tool for the job here. The nature of the holey landscape here is quite obscure, and should be sketched out more for the reader. For example, how much potential is there for mutational robustness to evolve given your set-up? Orr’s model is based on pairwise interactions–does your model have higher-order epistasis (I would imagine so, to some extent), and how might that affect our results? On the technical side, population sizes are tiny, necessitating high mutation rates. While this is somewhat understandable when using a computationally expensive model, these choices do increase the chances of double-mutations, maybe greatly accelerating the accumulation of DMIs. Certainly, the role of drift is magnified by this small range of population sizes. I feel that the downsides of this model are substantial, if indeed it limits you to these population sizes, and the upsides haven’t been fully realized in this paper.

There’s a lot to like here, and I think this paper can be a great contribution. I would like to see some response to these points, potentially leading to a little discussion of the limits and characteristics of this complex model.

Notes

Lines 11-12: I would turn this around–variation in build-up of RI should correlate with rates of speciation, but there are many other possible contributors.

Eq. 1–You describe this as “pairs of alleles,”, but I think you want to talk about polymorphic loci.

Lines 55-57: I don’t follow why you develop the intro in terms of DMIs, then switch to another focus at the very end of the Introduction? I feel like this paragraph either needs to be moved to Methods, where you can get into the technicalities, or motivated more clearly in the Intro.

Figure 2 is really nice!

Lines 108-112: I think you need to say a little bit to explain and justify this fitness function. Intuitively, the nature of selection and the shape/size of the neutral space are clearly quite important to the results. If you’re not going to vary its structure or parameters within this paper, we’re going to need a really good reason to believe that such variation is not important.

Lines 206-207: I would call this an “adaptive walk” if the steps are fitness-biased.

Figure 3(a): I think the changes in the intercept here with N should really be addressed in the Results text.

Reviewer #2: In computational studies of genotype to phenotype maps, it is common to introduce a fitness for each phenotype because that allows one to tackle (in silico) questions about populations and their evolution on such landscapes. Mutation-selection-drift (and recombination) balance leads to population steady state properties that are quite intuitive and have been mathematically justified in a number of limiting cases. In particular, the smaller the population, the lower the robustness (to mutations). Furthermore, in the presence of recombination, the population also becomes more robust to recombination (thus deleterious epistasis and in particular Dobzhansky–Muller incompatibilities are anti-selected). The present paper takes such a computational framework (based on the genotype to phenotype map when considering predicted RNA secondary structures) to probe the frequency of incompatibilities, motivated by its relevance in allopatric speciation. That motivation led the authors to simulate two independent populations and follow the dependence of the frequency of inter-population incompatibilities on the degree of genotypic "divergence", whereas most works (focusing on robustness, evolvability etc) consider a single population. The authors find that the previously mentioned properties hold as in a single population subject, namely (1) small population sizes lead to lower robustness and (2) recombination leads to a lower rate of deleterious epistasis. They also are able to address the dependence of incompatibility with genotypic divergence (neither linear nor quadratic in this model). From most theorists' point of view, the first two types of results given are exactly as expected while the last is novel as far as I know. My assessment is that the "originality" of the present work relies on the use of two populations. I note however that in the large population limit, inter and intra population observables are undistinguishable (a mathematical result that is rather easily shown). Thus this work's originality is confined to the cases of small population sizes.

Overall, the authors have performed a relatively sound computational work but I do object to their fixing the exponent c in Eq 4 by considering the averaging over N: indeed that exponent is different at N=1 and at N=infinity so it makes no sense to do any associated averaging. Unfortunately for the authors, if they accept my objection, the computations of Figs 3 and 4 have to be redone from scratch. Another weakness of their work is that all the simulations are done for RNA molecules of N=100 bases; mathematically, the definition of the snowball exponent c requires one take N to infinity. From the limited simulations presented it is not possible to infer whether the estimates provided for this snowball exponent are meaningful or reliable. The authors may circumvent this difficulty by for instance introducing a measure of snowballing that does not depend on a parametrization such as the one in Eq.4.

With respect to the issue of originality/novelty of the work, I do see some reasons to be positive but I do not agree with the authors' claim (cf. end of abstract) that they have provided new insights into mechanisms driving speciation.

Detailed points

- The title, abstract and even Fig 1 put forward "holey" landscapes but at the end of the introduction one finds out that it is just a neutral network. I found that way of presenting misleading and recommend they use the term neutral network throughout.

- The authors include a section covering "within" vs "between" population effects. Now that they know that in the limit of large population size these two cases are in fact identical (I invite you to include the proof to strengthen the paper), they can streamline this section and improve its scientific content.

- I don't understand why you put your study of incompatibilities in Drosophila within the Discussion section rather than have an associated Results subsection.

Reviewer #3: This paper argues that large populations with high rates of recombination (high N and high r) will evolve less postzygotic isolation for a given level of genomic divergence. Two distinct reasons are discussed (although they are first clearly distinguished only on lines 292-5)

(A) High-N and high-r populations experience more effective selection for genetic robustness (defined here as tolerance of new mutations). All else being equal, genotypes that are more tolerant of new mutations will also be more tolerant of introgressions at divergent sites. Almost by definition, introgressions into robust genotypes are less likely to cause incompatibilities. (Figs 3-4)

(B) High-r populations are less likely to diverge at sites that are involved in incompatibilities. This is because the incompatibilities may be expressed within populations, slowing the divergence (Fig. 5)

These predictions are compared to comparative evidence from the speciation literature, including two new comparative analyses, which suggest that Drosophila clades with higher mean Ne accumulate RI more slowly with genetic distance (Fig. 7), and have more fitness asymmetry in the F1 (Fig. 8).

This is a creative and intriguing paper, but I have some concerns about the presentation and the plausibility of the two mechanisms discussed.

Comments/questions about (A):

1. (A) posits that levels of mutational robustness will vary strongly with Ne and r. I worry that the strength of the effect in the simulations might be an artefact of neglecting variation in fitness among viable genotypes, so that selection for robustness is the only sort of selection that can take place. If other fitness variation were included, I wonder whether populations will evolve genotypes that are intrinsically fitter, regardless of their robustness. Data from Drosophila suggest that “speciation genes” involved in DMIs often diverged by positive selection.

2. Are there within-population data that might test the plausibility of (A)? For example, in the model the lethal mutation rate of a genotype is U(1-nu), so the model predicts a large increase in the lethal rate in low recombining regions of the genome, and in low-Ne species. E.g., I think that model predicts that the lethal rate will be much higher in the low-Ne D. sechellia than the high-Ne D. melanogaster. Are there any relevant data here?

Comments/questions about (B):

3. I was unsure how robust prediction (B) would be to variation in the parameters, or the fitness landscape. Previous work by Gavrilets, Fraisse and others suggests that the presence of potential incompatibilities does not slow down divergence by very much in many cases, and that rates of divergence will depend more on the intrinsic fitness differences between alleles (which are neglected in these simulations). I would expect the effects of r to also vary with NU. I think that the quantitative treatment, could be much fuller here.

4. Rather than the test shown in Figure 5, I think a simpler test and quantification of (B) would be to compare the realized proportion of introgressions that cause incompatibilities, to the expected proportion if sites had diverged at random (which I think is just (1-nu)).

Wider comments/concerns:

5. The fitness landscape:

The simulations use a fitness landscape based on RNA secondary structure. The properties of this landscape seem important to the results, but are difficult for the reader to understand or compare to previous data and theory (e.g. by Gavrilets). It is not obvious, for example, how many incompatibilities are effectively pairwise (involving 2-sites). Also unclear are the correlations in fitness between neighboring genotypes, the number of paths between fit genotypes etc. I am not sure if these stats are reported in previous work, but in either case, it would be useful to report them here.

My guess is that the computational burden of calculating secondary structure explains why simulations are restricted to very small populations, and to fixed values of certain parameters (e.g. alpha=12). It is therefore not clear to readers how robust the results are. I wonder whether simpler landscapes with similar properties could be used to extend the results, and provide a fuller exploration of parameter space?

I also wonder why small fitness differences were removed (in eq. 2). If the fitness function is slow to calculate, and prevents an analytical treatment, why use only 0 and 1 fitness values?

6. The relevance of the simulations to speciation data

Simulations use a haploid genotype of 100 sites assumed to represent a single molecule, and a small range of parameters (very small N etc), and often asexuals. Is this appropriate to the empirical work cited, which uses whole genomes of Drosophila, Solanum, Mus etc.?

Minor comments/questions:

7. The correlation in Fig. 7 looks impressive, but I wonder whether it is legitimate to ignore variation in Ne between species within each clade. E.g. the melanogaster group includes very low Ne species (e.g. sechellia) and high-Ne cosmopolitan species (melanogaster). It was not clear to me whether, if (A) and (B) are important, the rate of accumulation of RI will vary with the average Ne of the species in the clade.

8. Re. the title, is effect (A) best described as an effect of drift, or does it depend on the

population mutation rate, i.e. do results depend on NU or on N independently?

9. A lot of the past literature cited concerns differences in the rate of evolution of RI that are due to differences in the rate of evolutionary change. But the results here concern differences at a given level of genomic divergence. This important point could be made clearer.

10. The discussion of the snowball effect is interesting but seems peripheral to the main message of this paper. Also, a semantic point, should 1<c<2 as="" be="" described="" for="" orr="" support="">=2?</c<2>

**Have all data underlying the figures and results presented in the manuscript been provided?**

Reviewer #1: Yes

Reviewer #2: Yes

Reviewer #3: Yes

PLOS authors have the option to publish the peer review history of their article (what does this mean?). If published, this will include your full peer review and any attached files.

Reviewer #1: No

Reviewer #2: **Yes: **Olivier C. Martin

Reviewer #3: No

---

## [Decision Letter · Decision Letter 1]

6 Jan 2024

Dear Dr Azevedo,

We are pleased to inform you that your manuscript entitled "Genetic drift promotes and recombination hinders speciation on holey fitness landscapes" has been editorially accepted for publication in PLOS Genetics. Congratulations!

As you will read in the comments to the authors below, the reviewers were impressed by the content added to the revised version, and have only a few minor comments for you to consider, at your discretion, prior to uploading the final version.

Yours sincerely,

Lindi Wahl

Guest Editor

PLOS Genetics

Gregory Barsh

Editor-in-Chief

PLOS Genetics

Comments from the reviewers (if applicable):

Reviewer's Responses to Questions

**Comments to the Authors:**

Reviewer #1: This is an impressive revision that eliminates all the major issues I had, and goes well beyond that to produce a pretty comprehensive treatment of the topic. I think comparing the RNA model to the Russian roulette model was a good choice. The figures are clear and, reading it fresh, the questions that arise for me are generally answered well by subsequent results and discussion. The Intro also seems much improved. I can see that all reviewers raised somewhat different points and that it was a sizable task to respond to all of them. The authors should be commended for the effort. I think this is a really good paper now and should be accepted.

Reviewer #2: The authors have added much material. I appreciate especially the Russian roulette part. I no longer have any objections.

Reviewer #3: First, I apologize to the authors for the late review.

I commend the authors for the thoroughness of their revisions. They have made major changes, which deal with most of the concerns of all reviewers. The revised paper is both substantially different and substantially improved. Re. my previous point R3.15, this was sent the authors in garbled form. However, the concern (about what value of "c" counts as "snowballing") has been dealt with in the new discussion of the snowball effect.

I have two very small additional suggestions:

1. Re. my earlier point R3.3, I accept that data are not available, but if the authors agree that their model makes a prediction about the lethal mutation rate and Ne, this interesting prediction should probably be mentioned.

2. Re. my earlier point 3.4. Fraısse et al. 2014 JEB argued that the presence of DMIs did not greatly reduce the speed of neutral divergence. If possible, I would be interested to know how this claim relates to the present results (e.g. whether the conclusions of both sets of authors remain valid, or whether parameter regimes are different etc.).

**Have all data underlying the figures and results presented in the manuscript been provided?**

Reviewer #1: Yes

Reviewer #2: Yes

Reviewer #3: Yes

PLOS authors have the option to publish the peer review history of their article (what does this mean?). If published, this will include your full peer review and any attached files.

Reviewer #1: No

Reviewer #2: **Yes: **Olivier C. Martin

Reviewer #3: No

**Data Deposition**

http://datadryad.org/submit?journalID=pgenetics&manu=PGENETICS-D-23-00354R1

**Press Queries**

---

## [Editor Report · Acceptance letter]

17 Jan 2024

PGENETICS-D-23-00354R1 

Genetic drift promotes and recombination hinders speciation on holey fitness landscapes 

Dear Dr Azevedo, 

We are pleased to inform you that your manuscript entitled "Genetic drift promotes and recombination hinders speciation on holey fitness landscapes" has been formally accepted for publication in PLOS Genetics! Your manuscript is now with our production department and you will be notified of the publication date in due course.

With kind regards,

Anita Estes

PLOS Genetics

On behalf of:
